# Hospitalization costs and out-of-pocket (OOP) payment in lung cancer patients in Iran: Health Sector Evolution Plan (HSEP) has reduced OOP payments and improved financial protection

**Habib Jalilian[1,2], Somayeh Heydari[2], Elnaz Javanshir[3], Khosro Jamebozorgi[4], Nazanin Mir [2,5]\*, Abbas Eshraghi[6], Saeedeh Fehresti[7]**

1 Department of Health Services Management, School of Health, Ahvaz Jundishapur University of Medical Sciences, Ahvaz, Iran, 2 Iranian Center of Excellence in Health Management, Tabriz University of Medical Sciences, Tabriz, Iran, 3 Cardiovascular Research Centre, Tabriz University of Medical Sciences, Tabriz, Iran, 4 Faulty of Medicine, Zabol University of Medical Sciences, Zabol, Iran, 5 Health Management and Economics Research Center, Iran University of Medical Sciences, Tehran, Iran, 6 Student Research Committee, Tabriz University of Medical Sciences, Tabriz, Iran, 7 Department of Health Economics and Management, School of Public Health, Tehran University of Medical Sciences, Tehran, Iran

\* nazanin.mir2015@gmail.com

**Data Availability Statement:** Due to ethical restrictions and the involvement of a third-party

## Abstract

### Background and objective

In Iran, Health Sector Evolution Plan, the most significant reform in the healthcare system in recent decades, has been launched since 2014 with the objective of achieving universal health coverage, decreasing out-of-pocket health expenditures and improving access to health services in hospitals and clinics affiliated to the Ministry of Health and Medical Education (MOHME). This study aimed to estimate the hospitalization costs of lung cancer and the impact of HSEP on hospitalization costs of lung cancer and patients' contribution in Iran between 2010 and 2017.

### Methods

This was a prevalence-based cost of illness study with a bottom-up costing approach. The sample size included 1778 lung cancer patients hospitalized in the Imam Reza hospital in Tabriz, Iran, between May 5, 2010, to May 5, 2014, and four years after the implementation of Health Sector Evolution Plan: from May 5, 2014, to May 5, 2017. The analysis was conducted from a societal perspective. Data were extracted from the electronic medical records of patients and were analyzed using SPSS $_{V22.0}$, STATA $_{V13.0}$ and Microsoft Excel 2016. The Interrupted Time-Series design was applied to estimate the impact of the implementation of HSEP on hospitalization costs and patient contribution rate for reimbursement of costs.

organization, access to the data is subject to approval by the Corresponding Author and the Ethics Committee of the Tabriz University of Medical Sciences. Interested researchers can contact the Ethics Committee at info@tbzmed.ac.ir or by phone at +98413335 5926.

**Funding:** The author(s) received no specific funding for this work.

**Competing interests:** The authors have declared that no competing interests exist.

**Abbreviations:** HSEP, Health Sector Evolution Plan; OOP, out-of-pocket; COI, cost of illness; DAMA, Discharge against Medical Advice; AMA, against Medical Advice; MOHME, Ministry of Health and Medical Education; ITS, Interrupted Time-Series.

## Results

The mean hospitalization costs of lung cancer before and after the implementation of Health Sector Evolution Plan was estimated at 2860 ± 4575 and 5300 ± 8880 PPP (Current International $), respectively. Moreover, the amount of out-of-pocket payments reduced from 705 PPP (Current International$) (22.16%) before the implementation of Health Sector Evolution Plan to 480 PPP (Current International $) (10.5%) after its implementation. the hospitalization costs went up moderately before the HSEP (increased from 2320 $ in 2010 to 3025 $ in 2013). After the HSEP, it continued to rise, but with a more significant increase until 2016. Then, in 2016, it reached a peak (6395 $) before dropping in 2017 (5005 $). Regarding patient contribution, before the HSEP, the percentage of patient contributions increased from 19.45 in 2010 to 24.28 in 2013. With HSEP's implementation, this fell dramatically to 14.47 and continued to decline, reaching 7.99% in 2016. In 2017, patient contribution increased again and reached 9.58%.

## Conclusion

Overall, hospitalization costs experienced an upward trend over the course of study, but this trend considerably intensified further after the HSEP. The patient contribution demonstrated an upward trend before HSEP, followed by a significant decline post-HESP, and the percentage of out-of-pocket payments reduced after implementation of HSEP. Therefor this plan has been successful in achieving the goal of financial protection of patients.

## Background

Cancer is the second leading cause of death worldwide. Approximately 1 in 6 deaths are owing to cancer [1]. The number of new cases is expected to increase by nearly 70% over the next two decades [2]. Lung cancer accounts for the second most commonly diagnosed cancer among adults and 25% of all cancer deaths [3–7]. In 2020, lung cancer was the second most common cancer (2.21 million cases) and the leading cause of cancer death (1.8 million deaths) worldwide [1]. In 2020, lung cancer was the most common cancer in men worldwide, making up 15.4% of the total number of new cases diagnosed [8]. In 2020, the incidence and mortality rates and 5-year prevalence of lung cancer in both sexes were highest in Asia and lowest in Africa [9]. It has been estimated that the number of lung cancer worldwide will increase from 1.6 million in 2012 to 3 million in 2035 [10]. In Iran, lung cancer is one of the most common cancers and the second cause of cancer mortality [11]. In 2020, the numbers of new cancer cases, deaths, and prevalent cases (5-year) were estimated at 131191, 79136, and 319740, respectively. Moreover, the numbers of new cases, deaths, and prevalence cases (5 years) of lung cancer were estimated at 10465, 9071, and 11703, respectively [12].

The economic impact of cancer is substantial, as well [13]. Lung cancer is associated with a substantial economic burden in terms of both direct and indirect costs, representing 15 to 23% of the total cancer-related losses [14–17]. As the population ages, lung cancer treatment expenditure will become even more burdensome to the entire society [18]. Moreover, the economic burden of lung cancer differs across countries depending on the economic development level, healthcare systems, and purchasing power [18, 19]. In Iran, in 2017, the mean direct medical costs of lung cancer were estimated at 36637.02 PPP (Current International $), and the patient's contribution to the direct medical cost reimbursement was 2841.26 US$ (37.6% of the direct medical cost) [20].

Hospitals, as one of the essential components of the health systems, account for a large share of health sector expenditures. The cost of hospital services has considerably risen in the last decades and in most countries, more than 60% of the health sector expenditures are related to hospitals [21, 22]. Overall, there are a total of 924 hospitals in Iran, 61.68% of which are affiliated with the MOHME. Besides, hospital systems in the country, owing to a centralized political system, are similar in terms of policy and regulation, organizational structure, administrative and service delivery process, and degree of autonomy. Public hospitals in Iran have three main funding sources: 1) a line-item governmental budget (through the MOHME) and medical universities) and 2) reimbursement by Health Insurance Organizations, based on fee-for-service (FFS) and per-diem payments [23], 3) patients' contribution (out-of-pocket (OOP) payments) by copay, coinsurance, and deductible.

Iran's health system faces many challenges, like high OOP payments [24]. In the 5[th] development plan (2011–2015), 11% of the policies were related to the health sector emphasizing healthy humans and a comprehensive health approach. The policies were as follows: integration of policy making, planning, evaluation, developing quantity and quality of health insurance, supervision, public financing, and reducing OOP expenditures for health services to 30% by the end of the 5th plan [25]. According to the Eastern Mediterranean Regional Office (EMRO) report of the WHO, in Iran, in 2012, the share of OOP as % of Current Health Expenditure (CHE) was estimated at 57.52%. This high share of out-of-pocket payments indicates that a high portion of the expenditure has been imposed on households instead of insurance organizations [26]. However, despite various policies and plans of the MOHME to reduce OOP payments, this trended upward and eventually reached 52% in 2014 [27]. Therefore, on 5 May 2014, the Health Sector Evolution Plan (HSEP) was launched in Iran's health system to increase the financial protection of citizens against catastrophic healthcare expenditures and decrease hospitalization costs and the amount paid by the patients to <10%. It also aimed to improve access to health services in hospitals and clinics affiliated with the MOHME [28]. The HSEP has increased the annual healthcare system budget (an increase of 59% in 2015 compared to 2014) through the targeted subsidies law (10% of total subsidies) and 1% of value-added tax [28, 29].

Although various studies have been done to assess the effect of HSEP from various aspects, there have been no comprehensive studies on the effect of the HSEP on hospitalization costs associated with lung cancer patients in Iran. In 2017, the mean direct medical costs of lung cancer were estimated at 36637.02 PPP (Current International $), and the patient's contribution to the direct medical cost reimbursement was 2841.26 US$ (37.6% of the direct medical cost) [20]. Given that cancer is the most costly disease, and cancer patients incur a significant share of the costs, this study aimed to estimate the once-admitted hospitalization costs of lung cancer and investigate the impact of the implementation of HSEP on hospitalization cost of lung cancer and patient contribution rate in Tabriz, Iran.

## Materials and methods

### Study design

This prevalence-based cost of illness study was conducted from a societal perspective using a bottom-up costing approach.

### Setting and participants

The sample size included 1778 patients who were diagnosed with lung cancer and hospitalized in the Imam Reza hospital (the comprehensive center of admission for cancer patients) in Tabriz between 2010 and 2017 (four years before the implementation of the Health Sector Evolution Plan: from May 5, 2010, to May 5, 2014, and four years after the implementation of

Health Sector Evolution Plan: from May 5, 2014, to May 5, 2017). Sampling was conducted using the census method.

## Data collection process

Data were directly drawn from the electronic medical records of lung cancer patients (billing sheet), which is available in the Hospital Information System (HIS) of the Imam Reza hospital. This hospitalis one of the most equipped educational, medical, and research centers in northwest Iran. Billing of patients encompasses cost items in detail (costs of visits, surgery, anesthesia, chemotherapy, etc). Total costs, insurance contributions, patient contributions, and government subsidy contributions are separately available in the billing sheet of electronic medical records patients. Data were accessed from 24 September 2022 to 24 November 2022 for research purposes.

## Unit costs

A detailed description of unit costs is presented in Table 1.

## Ethics statement

This study was approved by the Ethics Committee of Tabriz University of Medical Sciences (Reference Number; IR.TBZMED.REC.1401.556). This study did not entail interaction with research subjects. We had no access to any patient identifying information as part of the study. Permission was obtained from the authorities of the hospital prior to the study.

## Data analysis

Data were analyzed using SPSS software $_{V22.0}$, STATA $_{V13.0}$ and Microsoft Excel 2016. SPSS and EXCEL were used for cost analysis. First, demographic variables and hospitalization costs were reported through descriptive statistics (Frequency and Percent) and mean ± SD. Then, ANOVA and T-test were used to analyze socio-demographic variables and hospitalization costs. Besides, a T-test was applied to compare the difference between lung cancer hospitalization costs before and after the HSEP. Finally, the Pearson correlation coefficient was used to evaluate the correlation between the patient's age, the length of hospital stay, and hospitalization costs.

**Table 1. Cost categories and sources of applied unit costs.**

| Sector | Service / Goods | Data source | Units | Monetary values (unit costs) |
|---|---|---|---|---|
| **Surgery costs** | Operating room consumables & equipment, Operating room medication, physician (surgeon) work, anesthetic | Medical records | Relative value unit/ Current Procedural Terminology | Medical tariffs |
| **Diagnostic costs** | Electrocardiography (ECG), pathology, Radiology, Laboratory Tests | Medical records | Quantity | Reimbursement schedule |
| **Visit Costs** | Visit in hospital | Medical records | Quantity | Medical tariffs |
| **Medication costs** | Medications that are recorded with a separate title in hospital billing codes | Medical records | Quantity | Reimbursement schedule |
| **Hoteling costs** | Cost of non-physician human resources, depreciation, repairs and maintenance, food, energy, other goods and services not included in the billing separately, Hospital services and equipment, Nursing services | Medical records | Days of hospital stay | Reimbursement schedule |
| **Other hospitalization costs** | Intravenous chemotherapy cost, cost of faculty members, Forensic medicine cost | Medical records | Quantity | Reimbursement schedule |

In our study, the Interrupted Time-Series (ITS) design was applied to estimate the impact of the implementation of HSEP on hospitalization costs and patient contribution rate for reimbursement of costs. ITS analysis is considered as one of the strongest quasi-experimental research designs and is useful when a randomized trial is not feasible or unethical [30, 31]. An ITS study does not require a concurrent "control group" to establish a causal link between an intervention and an outcome [30, 32]. An ITS is used when examining whether the data pattern observed post-intervention differs from that observed pre-intervention. The ITS analysis is a valuable study design for evaluating public health interventions; it is particularly suited to interventions introduced at a population level over a given time period and target population-level health outcomes [33].

Our sample included 92 observations (i.e., one per month), starting 46 months before the HSEP and 46 months after the HSEP. In this model, baseline measurements of the dependent variables are made periodically before and after the intervention.

A segmented regression model was applied to analyze the Time-Series data. This model reveals the regression model by Ordinary Least Squares (OLS) regression. In time series analysis, changes in the dependent variables because of interventional variables are divided into two sections: change in level and change in slope. A change in level represents a short-run change; a change in slope represents a long-run change in the dependent variable. In this model, stationary, autocorrelation, and variance heterogeneity were tested using STATA software.

In order to reduce the effect of inflation and increase the comparability of costs in different periods, all costs in this study were adjusted for PPP (Current International $) and discounting was conducted. Most of the literature has used a discount rate of 3%, but in this study, hospitalization costs were discounted at four different discount rates (3%, 5%, 7%, and 10%).

## Results

The socio-demographic characteristics are shown in Table 2. The mean age of patients before and after the HSEP was estimated at 60 and 61 years of age, respectively. The estimated mean mortality age before and after the HSEP was 65.66 and 62, respectively. Moreover, the average length of hospital stay before and after HSEP was 8.17 and 8.3, respectively. In this study, 74.7% of the patients were men. Most patients were over 60 years old and covered by basic insurance. Only 6% of patients were covered by supplemental insurance.

In addition, 7.1% of patients had Discharge against Medical Advice (AMA) before completing the treatment process. There was a significant difference between discharge status before and after the implementation of HSEP. The percentage of DAMA after the implementation of HSEP decreased significantly compared with before HSEP (5.5% Versus 8.7%), and this difference was statistically significant (P < .0001).

According to Table 3, the total hospitalization costs of lung cancer after HSEP increased significantly compared with before HSEP ($5300 Versus 2860), and this difference was statistically significant (P < .0001). There was a significant difference between discharge status and the mean hospitalization costs of lung cancer patients. Hospitalization costs were lower in patients who were discharged against Medical Advice (AMA) (P < .0001). Besides, hospitalization costs had a significant positive correlation with the length of hospital stay (P < .0001, CC = 0.713) but did not have a significant correlation with age (P = 0.33, CC = 0.023).

Before HSEP, the main driver of hospitalization costs of lung cancer were medicine costs ($830), followed by hoteling ($825) and surgery costs ($530). On the contrary, after the implementation of HSEP, the main driver of hospitalization costs were hoteling costs ($1925), followed by surgery ($1000) and medicine costs ($920). In addition, the results showed that the proportion of both basic and supplemental insurance for reimbursement costs increased

**Table 2. Socio-demographic characteristics and total hospitalization cost of lung cancer.**

| Variables | Modes | Frequency (%) | Before implementation of HSEP | | After implementation of HSEP | |
|---|---|---|---|---|---|---|
| | | | Mean± SD | P-value | Mean± SD | P-value |
| Gender | Male | 1329 (74.7) | 2840 ± 4460 | 0.84 | 5220 ± 8900 | 0.59 |
| | Female | 449 (25.3) | 2905 ± 4850 | | 5606 ± 8830 | |
| Age | < 40 | 145 (8.2) | 3030 ± 2790 | 0.92 | 4190 ± 3170 | 0.56 |
| | 40–60 | 707 (39.8) | 2865 ± 5580 | | 5515 ± 6765 | |
| | > 60 | 926 (52.1) | 2825 ± 3960 | | 5285 ± 10660 | |
| Basic insurance status | Yes | 1762 (99.1) | 2870 ± 4580 | 0.46 | 5320 ± 8900 | 0.07 |
| | No | 16 (.9) | 1850 ± 4000 | | 1850 ± 3209 | |
| Type of basic insurance | Social security insurance | 611 (34.4) | 2460 ± 3420 | 0.005 | 5108 ± 6825 | 0.54 |
| | Armed force insurance | 100 (5.6) | 4398 ± 12070 | | 3670 ± 4335 | |
| | Iranian health insurance | 979 (55.1) | 2820 ± 3780 | | 5625 ± 10284 | |
| | Relief foundation insurance | 50 (2.8) | 3370 ± 4495 | | 5000 ± 5640 | |
| | Other basic insurances | 38 (2.1) | 4965 ± 5400 | | 3100 ± 3460 | |
| Supplemental Insurance status | Yes | 107 (6.0) | 3160 ± 3280 | 0.48 | 5490 ± 6608 | 0.88 |
| | No | 1670 (94.0) | 2840 ± 4655 | | 5290 ± 8995 | |
| Discharge status | discharge | 1328 (74.7) | 2450 ± 2845 | < .0001* | 4765 ± 8445 | < .0001* |
| | Death | 323 (18.2) | 4920 ± 8345 | | 8485 ± 11000 | |
| | self-discharge | 127 (7.1) | 1607 ± 2285 | | 3230 ± 4580 | |

*P < 0.05 was considered as significant
ANOVA and T-test used.

significantly after HSEP. Before the implementation of the HSEP, 80% of costs were paid by basic insurance, while after the HSEP, 89% of costs were paid by basic insurance, and this difference was statistically significant. Moreover, the proportion of supplemental insurance for reimbursement of costs increased significantly from $3.87 before the HSEP to $44.03 after the HSEP. Before HSEP, government subsidy contribution was zero, but after the HSEP, it reached

**Table 3. Mean ±SD of Hospitalization cost of lung cancer before and after the HSEP.**

| Costs items | Before | After | P-value |
|---|---|---|---|
| Surgery | 530 ± 1240 | 1000 ± 2470 | < 0.001* |
| Diagnosis | 280 ± 650 | 595 ± 850 | < 0.001* |
| Visit | 160 ± 208 | 350 ± 440 | < 0.001* |
| Hospital services and equipment | 160 ± 320 | 260 ± 540 | < 0.001* |
| Medicine | 830 ± 2205 | 915 ± 2215 | < 0.001* |
| Nursing services | 50 ± 80 | 105 ± 200 | < 0.001* |
| Hoteling | 825 ± 1450 | 1925 ± 4730 | < 0.001* |
| Other hospitalization costs | 35 ± 55 | 185 ± 270 | < 0.001* |
| Total hospitalization cost | 2860 ± 4575 | 5300 ± 8880 | < 0.001* |
| Basic insurance contribution | 2290 ± 35100 | 4755 ± 8305 | < 0.001* |
| Supplementary insurance contribution | 4 ± 67 | 44 ± 323 | < 0.001* |
| Government subsidy contribution | 00 | 460 ± 870 | < 0.001* |
| Patient contribution | 705 ± 1510 | 480 ± 880 | 0.59 |

*P < 0.01 was considered as significant
U Mann Whitney test used.

$460, and this difference was statistically significant. Besides, compared with before the HSEP, patients' contribution was reduced after the HSEP (22.16% Versus 10. 5%), but this difference was not statistically significant.

As shown in Fig 1 and Table 4, before adjusting for PPP (Current International $), the hospitalization costs went up moderately before the HSEP. After the HSEP, it continued to rise, but with a more significant increase until 2016. Then, in 2016, it reached a peak before dropping in 2017. After adjusting for PPP (Current International $), before the HSEP, from 2010 to 2013, costs fluctuated steadily. However, after the implementation of HSEP, costs went up considerably and continued to rise for three consecutive years, peaking in 2016. After that, it saw a considerable decline in 2017 (Fig 2).

Before the HSEP, the percentage of patient contributions increased from 19.45 in 2010 to 24.28 in 2013. As with HSEP's implementation (2013), this fell dramatically to 14.47 and continued to decline, reaching 7.99% in 2016. In 2017, patient contribution increased again and reached 9.58% (Fig 3). Additionally, no costs were covered by government subsidies before the HSEP. By implementing HSEP in 2013, 6.29% of costs were covered by government subsidies. In 2015, it rose 11.02% before decreasing to 8.7% in 2016 and 7.91% in 2017.

The results of the segmented regression model are presented in Table 5. The starting level of total hospitalization costs was 7543.91$. The hospitalization costs rose by 403.88$ every month before the intervention (HSEP). The increase in the total hospitalization cost was statistically significant (*P*-value for baseline trend < 0.01). A statistically significant increase in hospitalization costs of 13866.67 (P = 0.01, 95% confidence interval = 2309.17–25424.17) was observed in the first month after the intervention. In addition, no significant change was observed in the monthly trend of hospitalization costs (compared with the period before the intervention) (P = 0.13, 95% CI = -762.4–292.8). Segmented regression also indicated that after the intervention implementation, the total hospitalization cost increased monthly to 169.28, but this increase was not statistically significant (P = 0.49, 95% CI = -325.738–663.915). In addition, the starting level of the amount paid by the patient was estimated at 1157.73$. The

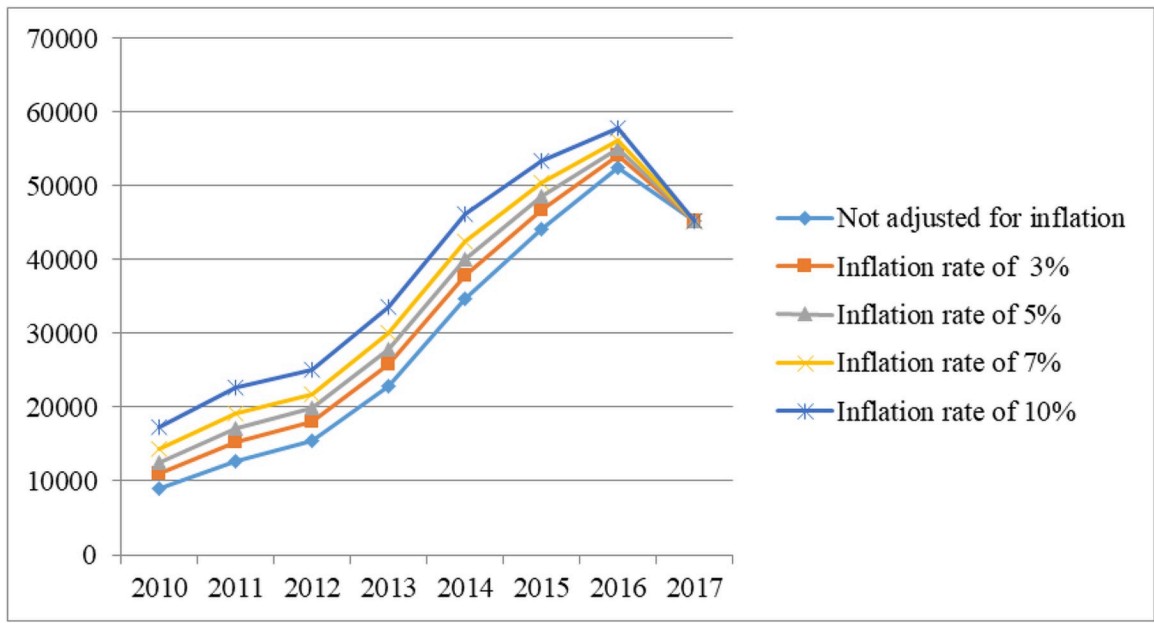

**Fig 1. Trend of hospitalization costs of lung cancer (Itran Rials).**

**Table 4. Mean± SD of hospitalization cost of lung cancer by study years and discounting.**

| Year | without discount rate | Discount rate of 3% | Discount rate of 5% | Discount rate of 7% | Discount rate of 10% |
|---|---|---|---|---|---|
| **Iran Rials (Thousands Rials)** | | | | | |
| 2010 | 8909 ± 11720 | 10957 ± 14414 | 12536 ± 16491 | 14306 ± 18819 | 17361 ± 22839 |
| 2011 | 12724 ± 26529 | 15231 ± 31748 | 17094 ± 35631 | 19144 ± 39902 | 22598 ± 47103 |
| 2012 | 15553 ± 31722 | 18030 ± 36774 | 19850 ± 404860 | 21814 ± 44491 | 25048 ± 51088 |
| 2013 | 22891 ± 23821 | 25764 ± 26811 | 27825 ± 28955 | 30006 ± 31225 | 33515 ± 34877 |
| 2014* | 34675 ± 57998 | 37891 ± 63376 | 40141 ± 67140 | 42479 ± 71050 | 46153 ± 77196 |
| 2015 | 44247 ± 45346 | 46798 ± 48083 | 48633 ± 49968 | 50503 ± 51890 | 53375 ± 54840 |
| 2016 | 52529 ± 106990 | 54105 ± 110199 | 55156 ± 112339 | 56206 ± 114479 | 57782 ± 117689 |
| 2017** | 45327 ± 71454 | 45327 ± 71454 | 45327 ± 71454 | 45327 ± 71454 | 45327 ± 71454 |
| **Adjusted in terms of Purchasing Power Parity[a] (current international $)** | | | | | |
| 2010 | 2320±3050 | 2850 ± 3750 | 3260 ± 4290 | 3725 ± 4900 | 4520 ± 5945 |
| 2011 | 2730±5695 | 3270 ± 6815 | 3670 ± 3670 | 4110 ± 8570 | 4850 ± 10115 |
| 2012 | 2740 ± 5590 | 3180 ± 6485 | 3500 ± 7140 | 3845 ± 7845 | 4415 ± 9010 |
| 2013 | 3025 ± 3150 | 3405 ± 3540 | 3680 ± 3825 | 3965 ± 4430 | 4430 ± 4610 |
| 2014* | 4210 ± 7040 | 4600 ± 7690 | 4870 ± 8150 | 5155 ± 8625 | 5600 ± 9370 |
| 2015 | 5405 ± 5540 | 5720 ± 5875 | 5940 ± 6105 | 6170 ± 6340 | 6520 ± 6700 |
| 2016 | 6395 ± 13030 | 6590 ± 13420 | 6720 ± 13680 | 6845 ± 13940 | 7035 ± 14330 |
| 2017** | 5005±7890 | 5005 ± 7890 | 5005 ± 7890 | 5005 ± 7890 | 5005 ± 7890 |
| **P value** | < .0001 | < .0001 | < .0001 | < .0001 | < .0001 |

ANOVA test used.

*Year of HSEP implementation

**Base year for discounting rate

a. World Bank, International Comparison Program database. PPP conversion factor, GDP (LCU per international $)

https://data.worldbank.org/indicator/PA.NUS.PPP?end=2017&start=199

figure increased by 117.74$ every month before the intervention (HSEP), and this decrease was statistically significant (*P*-value for baseline trend = < .0001). A statistically significant decrease in the amount paid by the patient of 2074.11 (P = 0.013, 95% confidence interval = -3701.20 –-447.01) was observed in the first month after the intervention. In addition, a significant decrease in the monthly trend of patient contribution (compared with the period before the intervention) was observed, and it was estimated at 149.44 (P = 0.00, 95% CI = -202.30 –-96.57). Segmented regression indicated that after the implementation of the intervention, the amount paid by the patient decreased monthly to 31.695, but this decrease was not statistically significant (P = 0.15, 95% CI = -75.962–12.571). Moreover, the percentage of patient copayment was estimated at 0.183 at the starting month of the study period. There was a significant increase in the proportion of patient copayment (P = < .0001, 95% CI = 0.000 to 0.002) during the period before the intervention. The estimated percentage of patient copayment decreased by 0.116% (P < .0001, 95% CI = -0.019 to -0.83) in the first month after introducing the HSEP. Also, we observed a significant decrease in the monthly trend of patient copayment after the introduction of the intervention (P < .0001). Segmented regression also indicated that after the implementation of the intervention, the patient copayment decreased monthly to 0.001, and this decrease was statistically significant (P = 0.003, 95% CI = -0.002 –-0.000. Overall, while the total hospitalization costs and the amount paid by the patient did not change significantly after the introduction of the intervention, the percentage of patient copayment decreased significantly.

Figs 4–6 visually display these results before and after the intervention.

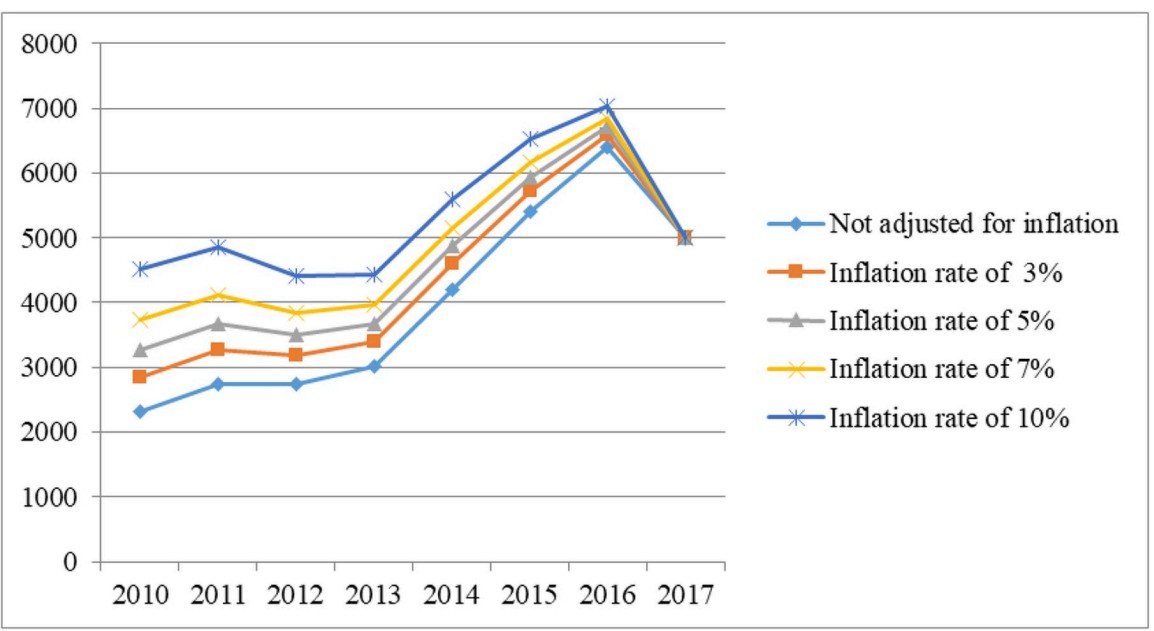

**Fig 2. Trend of hospitalization costs of lung cancer (After adjusting for PPP (Current International $)).**

Before the implementation of HSEP, the variation in hospitalization costs showed a slight scattering, but it experienced a further scattering after the HSEP. Also, costs saw a sudden increase after the HSEP and showed a rising slope throughout the study (Fig 4). Moreover, before HSEP, the amount of patient copayment had a relatively steep rise, while after HSEP, it dropped remarkably, and experienced a declining trend with a slight slope (Fig 5). On the other hand, according to Fig 6, before the HSEP, the percentage of patient copayment was high and had a rising trend, while after the HSEP, it fell and experienced a decreasing slope. After which, the figure saw a rise again in 2017.

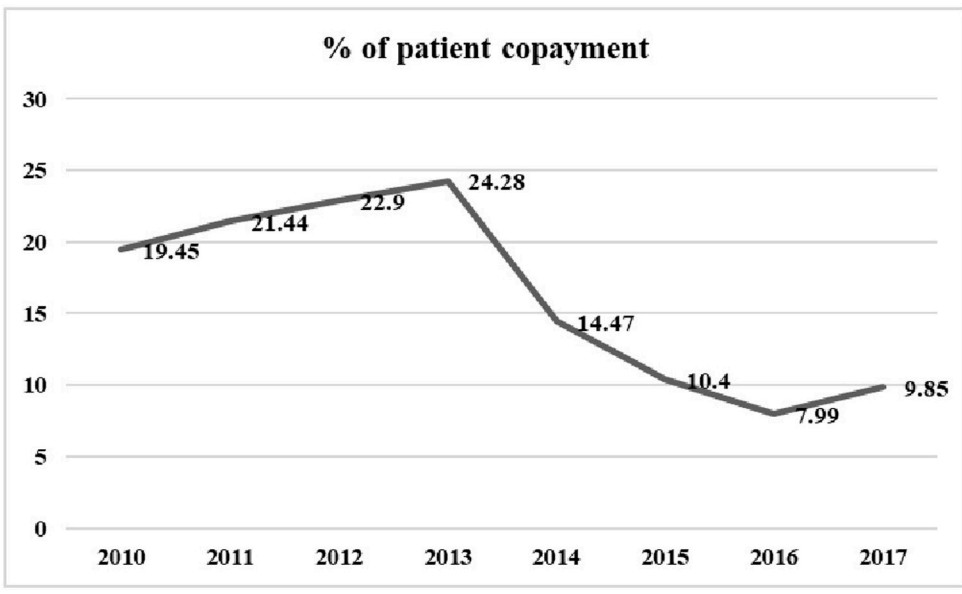

**Fig 3. Trend of patient contribution rate of hospitalization cost of lung cancer.**

**Table 5. Estimated coefficients of the segmented regression model for hospitalization cost, the amount paid by the patient, and patient contribution rate.**

Regression with Newey-West standard errors
maximum lag: 5 Number of observation = 96 F (3, 92) = 37.48 p-value = 0.0000

| Parameter (Variables) | Coefficient | SE [a] | t | P-Value | [95% Conf. Interval] | |
|---|---|---|---|---|---|---|
| | | | | | Lower | Upper |
| **Total hospitalization costs** | | | | | | |
| Intercept, (cons) $\beta_0$ | 7543.91 | 1696.2 | 4.45 | < .001 | 4175.2 | 10912.5 |
| Pre-intervention slope, $\beta_1$ | 403.88 | 63.66 | 2.3 | < .001 | 277.44 | 530.31 |
| Change in slope, $\beta_2$ | 13866.67 | 5819.23 | 6.34 | 0.01 | 2309.17 | 25424.17 |
| Change in trend, $\beta_3$ | -234.79 | 265.6 | -0.88 | 0.37 | -762.4 | 292.8 |
| Linear trend, $\beta_{p1}$ | 169.088 | 249.146 | 0.678 | 0.499 | -325.738 | 663.915 |
| Post-intervention linear trend [b] | | | | | | |
| **The amount paid by the patient** | | | | | | |
| Intercept, $\beta_0$ | 1157.73 | 377.69 | 3.07 | 0.003 | 407.58 | 1907.87 |
| Pre-intervention slope, $\beta_1$ | 117.74 | 14.55 | 8.09 | < .0001 | 88.84 | 146.64 |
| Change in slope, $\beta_2$ | -2074.11 | 819.24 | -2.53 | 0.013 | -3701.20 | -447.01 |
| Change in trend, $\beta_3$ | -149.44 | 26.61 | -5.61 | < .0001 | -202.30 | -96.57 |
| Linear trend, $\beta_{p1}$ | -31.695 | 22.288 | -1.422 | 0.158 | -75.962 | 12.571 |
| Post-intervention linear trend [b] | | | | | | |
| **Patient co-payment** | | | | | | |
| Intercept, $\beta_0$ | 0.183 | 0.012 | 15.29 | < .0001 | 0.1600 | 0.207 |
| Pre-intervention slope, $\beta_1$ | 0.0013 | 0.000 | 3.81 | < .0001 | 0.000 | 0.0021 |
| Change in slope, $\beta_2$ | -0.116 | 0.016 | -6.99 | < .0001 | -0.019 | -0.83 |
| Change in trend, $\beta_3$ | -0.002 | 0.000 | -4.72 | < .0001 | -0.0039 | -0.001 |
| Linear trend, $\beta_{p1}$ | -0.001 | 0.000 | -3.006 | 0.003 | -0.002 | -0.000 |
| Post-intervention linear trend [b] | | | | | | |

[a] Newey–West standard errors.

[b] This obtained from the following time trend equation: $Y_{pt} = \beta_{p0} + \beta p1 * time_{pt} + \varepsilon t$, where, $Y_{pt}$ is the value of total hospitalization costs, the amount paid by the patient and the percentage of patient copayment at time $t$ after the intervention and time $_{pt}$ is the time trend variable which takes values between 1 (first observation after the intervention) and 50 (last observation after the intervention).

## Discussion

This study was designed to estimate the hospitalization costs of lung cancer and the impact of HSEP on hospitalization costs of lung cancer and patients' contribution in Iran between 2010 and 2017. In this study, the total hospitalization costs increased significantly from $2860 before HSEP to $5300 after HSEP (an increase of almost 1.85 times). A study by Afkar et al. showed that the total means of hospitalization costs of breast cancer increased from 1490 PPP (Current International $) before implementing HSEP to 4345 PPP (Current International $) after its implementation [34]. In our study, after applying discount rates of 3%, 5%, 7%, and 10%, total costs after the HSEP rose significantly compared with before the HSEP. Besides, before the HSEP, the total hospitalization costs experienced an increase of 1.3 times without adjusting costs for discount rates, while after applying discount rates of 3%, 5%, 7%, and 10%, total hospitalization costs increased by 1.19, 1.12, 1.06, and 0.98, respectively (between 2010 and 2013). After the HSEP, the incremental rate of costs with the similar discount rates was 1.18, 1.02, 0.97, and 0.89, respectively (from 2014 to 2017). It was noticeable that whether applying the discount rate or not, the incremental rate before the HSEP was higher than after the HSEP. This might be due to the coincidence of the implementation of HSEP with severe political and economic sanctions, reduced oil sales and inflation, and economic hardship, which led to an

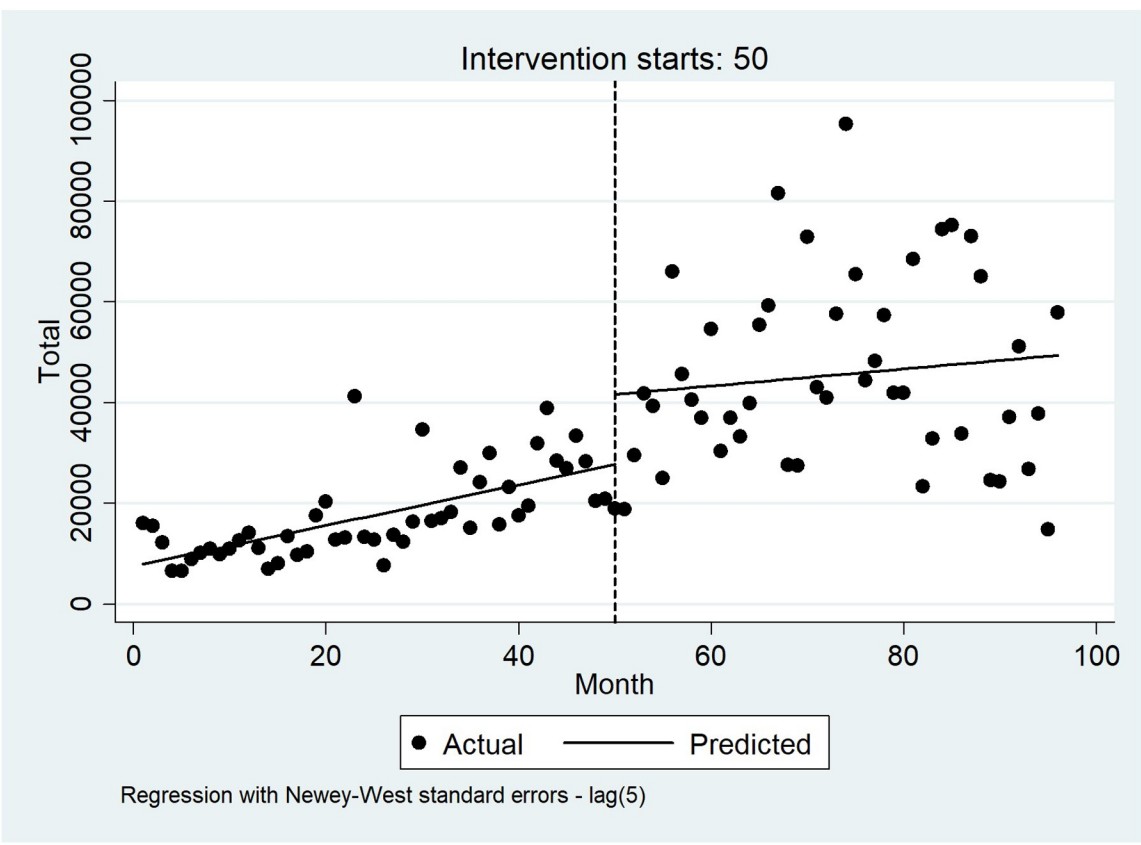

**Fig 4. Interrupted time series analysis with Newey–West standard errors and five lags for total hospitalizations costs.**

increase in healthcare costs. Patient financial burden increased in 2017 as an indirect consequence of the economic climate, but the HSEP prevented this from increasing even more than it would have done otherwise.

According to the findings, all components of hospitalization costs increased significantly after the implementation of HSEP. Before the implementation of HSEP, the highest hospitalization costs were medicine, hoteling, and surgery costs, making up 29.10%, 28.77%, and 18.54% of hospitalization costs, respectively. After the HSEP, the highest hospitalization costs were observed in the hoteling, surgery, and medicine costs, accounting for 36.28%, 18.82%, and 17.25%, respectively. The highest increase after the HSEP was observed in hoteling costs (an increase of 2.33%) and medicine costs (an increase of 2.33%). Interestingly, increased costs after the HSEP was more related to hospital services than treatment procedures. It is important to note that the HSEP mainly aims to improve hoteling services in educational hospitals and increase the level of payment to physicians working in these hospitals to encourage them to stay in public hospitals. Hence, hoteling costs, unlike medicine costs, have increased significantly.

During the eight-year study period, the mean hospitalization costs of lung cancer had an upward trend. This trend continued even by adjusting costs for a 10% rate and adjusting costs for PPP (Current International $). However, this increase had a steeper slope after HSEP implementation. This increase can be attributed partly to technological advances and more expensive medicine related to cancer treatment. Another reason may be that more financial resources are allocated to the health system, which is obtained from implementing the Subsidies Targeting Plan. The plan is mainly supported through an increased public annual budget for the health

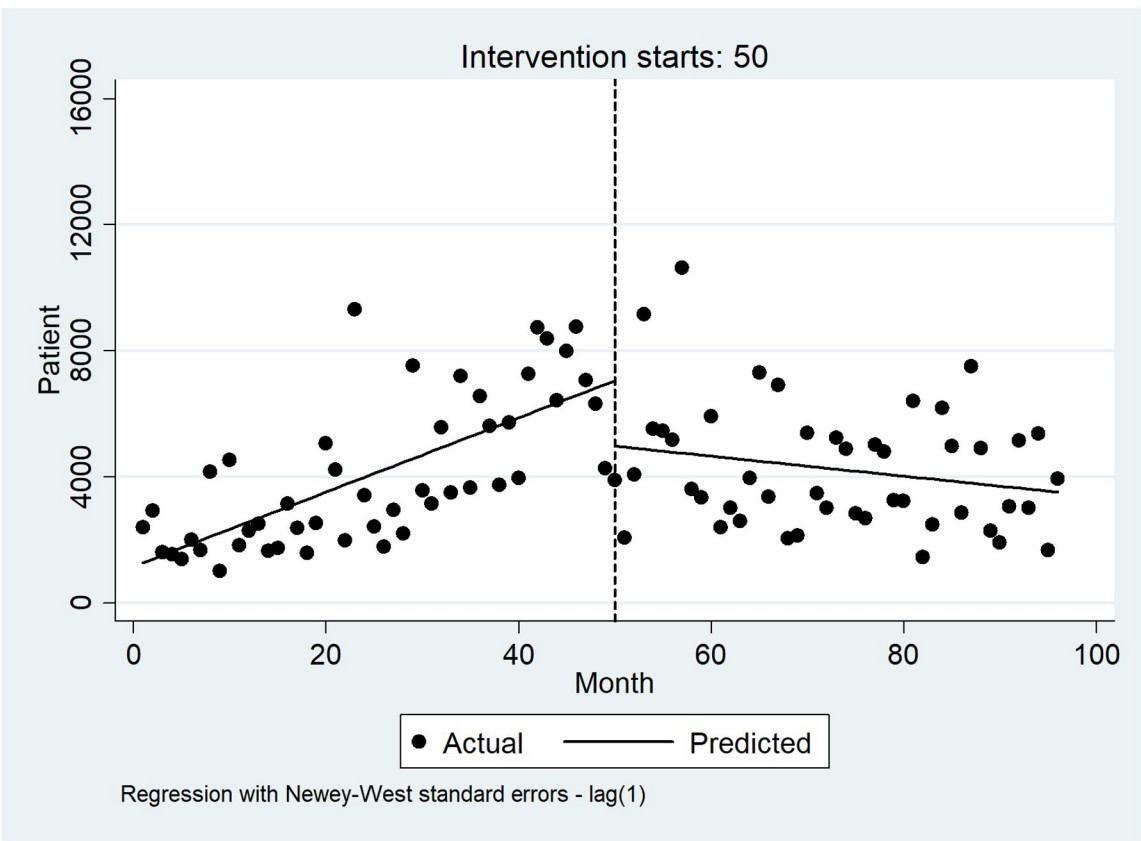

**Fig 5. Interrupted time series analysis with Newey–West standard errors and no lag for the amount paid by the patient.**

sector, resources of the targeted subsidies' law (10% of total subsidies), and a specific 1% value-added tax (VAT) for health. It has been estimated these resources led to approximately a 70% increase in health sector resources [28, 35, 36]. Also, part of the increase in cancer costs can be attributed to increased medical tariffs. The third phase of the HSEP began on Sep 29, 2014, which updated health services' tariffs in all parts of the health system [37]. However, since 2017, the hospitalization costs of lung cancer have declined because of falling oil prices and the intensification of political and economic sanctions, which shows HSEP has not been based on sustainable resources.

In this study, after the HSEP, the proportion of basic insurance, supplemental insurance, and government subsidy for reimbursement costs increased significantly. Before the implementation of the HSEP, 80.10% of costs were paid by basic insurance, but this rose to 89.69% after the HSEP. In our study, the reduction in the amount paid by patients was not statistically significant. Moreover, although the proportion of supplemental insurance for reimbursement of costs increased significantly after HSEP implementation, this accounts for a small part of hospitalization costs, indicating that private insurance in Iran's healthcare system has not yet developed sufficiently. Furthermore, Before the HSEP, no costs were covered by government subsidy. But after HSEP, 8.78% of costs were covered by government subsidy.

Although the percentage of patient contribution for reimbursement of cancer costs fell after the implementation of HSEP, the actual amount paid by a patient did not change significantly. This is because the total costs rose, and a patient paid only less percentage of a larger amount of costs. Another possible reason might be that although health expenditures had a rising

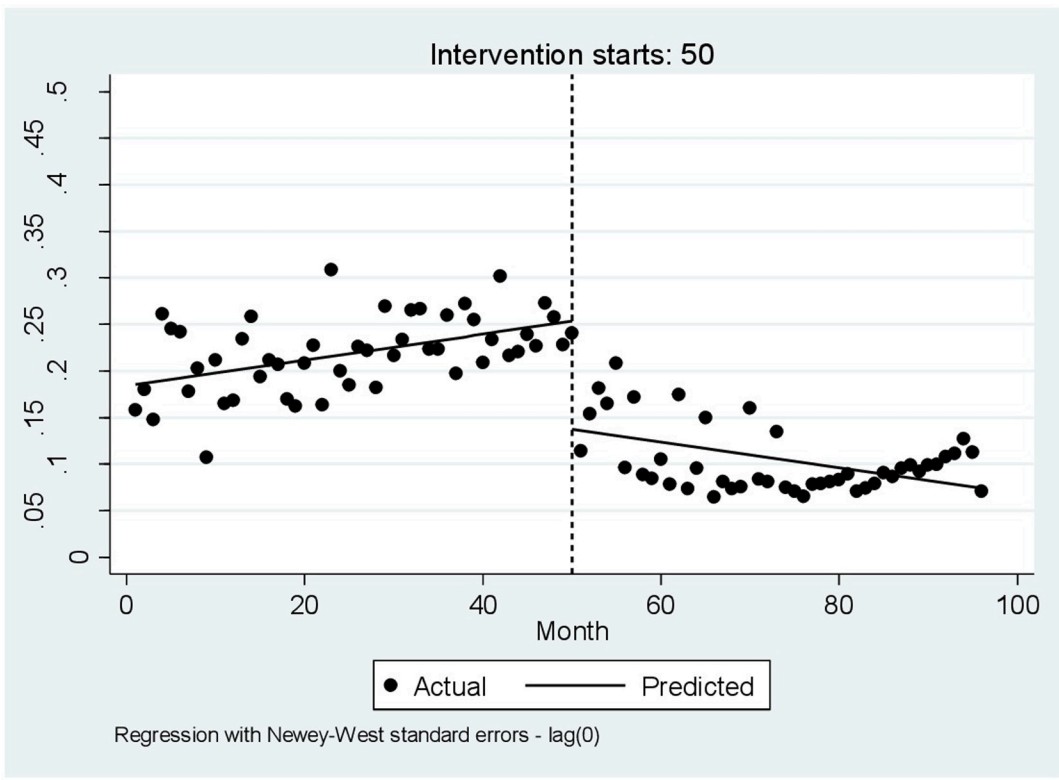

**Fig 6. Interrupted time series analysis with Newey–West standard errors and no lags for the patient copayment.**

trend due to inflation and sanction, the plan largely reduced the financial burden imposed on patients. Moreover, before HSEP implementation, the percentage of patient copayment had an increasing trend, while it showed a decreasing trend after HSEP implementation. However, due to falling oil prices and economic sanctions, this trend has increased since 2017, imposing an increased financial burden on the patient. In 2017, in Iran, the patient's copayment for the direct medical cost of lung cancer amounted to 2841.26 US$, equivalent to 37.6% of the direct medical cost (20). Another study in Iran indicated that the percentage of breast cancer patient co-payment decreased after the implementation of HSEP, whereas the total amount of patient contributions increased after its implementation, which is consistent with this study's results [34]. A study by Piroozi et al. (2017) demonstrated that the OOP payments declined after the implementation of HSEP in hospitals affiliated with the (MoHME) and in hospitals affiliated with the Social Security Organization (SSO) [37]. Another study suggested that the average OOP expenditures for cesarean were 16.05% before the implementation of HSEP, while it declined to 4.30% and 2.5%, respectively, after the HSEP [38].

The hospitalization costs of patients who were covered by health insurance were more than 2.5 times greater than those without insurance coverage. Part of this could be due to the induced demand resulting from insurance, indicating the financial unaffordability of unin-sured people to pay for illness and the withdrawal from some essential services. The hospitali-zation cost of cancer patients who died was about twice as high as those who were discharged and more than three times the cost of patients who were discharged against Medical Advice. Patients who die are likely to be in more advanced stages of the disease and are likely to need more complex and expensive care. Of course, the cost to these patients is ineffective and leads

to a waste of health system resources. Hence, it seems that palliative care can be useful among patients who have a low likelihood of being improved.

Although the cost for patients who had been DAMA is lower, they might have been deprived of the necessary services. Supportive measures should be taken for these patients if the patient's discharge is due to the disease costs and the lack of financial ability in cost payment. Decreased DAMA after the HSEP can be due to the increase in the quality of hotel services in educational hospitals, as well as the increase in the level of financial support for patients by reducing the contribution of patients. Improving the quality of hoteling services and increasing the level of financial support for patients were the goals of HSEP, and reducing the DAMA can be considered as one of the positive consequences of this plan for those who are DAMA or have to refer to the private sector and incur additional costs or forgo treatment process altogether. Tourani et al., in 2016, investigated the changes in the performance and quality of the Emergency Department (ED) after HSEP. Their results demonstrated a reduction of 1.11% in DAMA after HSEP [39]. Similarly, previous studies in Iran demonstrated that the rate of DAMA decreased after HSEP [40, 41].

Studies related to the cost of illness and economic evaluation can be performed from the following perspectives: 1) the health plan enrollee, 2) the patient, 3) the health plan manager, 4) the provider, 5) the technology manufacturer, 6) the specialty society, 7) government regulators, or 8) society as a whole [42]. Although the evaluation of this study can only confirm the plan's success from the perspective of patients, if it was evaluated from the opportunity cost approach and the perspective of society as a whole, it might provide different results. Therefore, it is suggested that future studies consider opportunity cost and societal perspective when evaluating the effectiveness of interventions and plans in the health sector.

Considering the limitation of the healthcare system resources, additional resources allocated to new interventions and programs or increased resources of previous interventions and programs cause some other potential health interventions to be inevitably forgone. Hence, we need to consider "opportunity cost" and, based on the Cost-effectiveness thresholds (CETs), assess whether interventions are worthwhile [43].

## Policy implications

According to the findings, the implementation of HSEP has failed to reduce patients' OOP payments and increase financial protection. It appears that some factors made the plan deviate from its initial objectives. Firstly, since the implementation of the HSEP coincided with the worsening of political and economic sanctions, a decline in oil prices and exports, and economic inflation, the financial sustainability of the plan faced multiple problems. Secondly, HSEP was financed by the targeted subsidies (a plan which removed the government subsidies from essential goods), and only part of this subsidy was allocated to the health sector. Besides, with a sudden increase in medical tariffs, these resources have led to an increase in the price of services and physicians' wages rather than an improvement in service quality and an increase in patients' financial protection. So, this plan was more in favor of physicians rather than patients. Therefore, it is necessary for health policymakers to comprehensively consider factors affecting the plan when designing and implementing reforms.

## Generalizability of the results

Iran's health system is a centralized system consisting of the Ministry of Health and 46 universities of Medical Sciences affiliated with it. Moreover, policy-making is as top-down across the country. Iran's Ministry of Health has the main authority and sets the same regulations for all universities in each province, and the universities are the executive representatives. Given that the regulations are established at a capital level and notified to provinces, there is no difference

between the provinces. Therefore, this hospital can be an appropriate representative regarding the extent to which of the effect of the implementation of the plan on healthcare expenditures.

## Limitations

Despite its strengths, the study has limitations worth considering. Firstly, outpatient costs were not examined due to the non-availability of information related to some patients and their caregivers over the course of this study. Secondly, this study focused on only some plan objectives and examined only its effect on hospitalization costs in lung cancer patients. Therefore, based on the results of this study, it is impossible to assess the plan's overall performance. Another limitation of our study was that we did not have a non-intervention control group.

## Conclusions

After the implementation of the HSEP, the hospitalization cost of lung cancer increased dramatically. This can be attributed to increased resources available in the health system and medical tariffs. By contrast, the percentage of OOP payments reduced after the HSEP. Patient financial burden increased in 2017 as an indirect consequence of the economic climate, but the HSEP prevented this from increasing even more than it would have done otherwise.

## Supporting information

**S1 Checklist. STROBE statement—checklist of items that should be included in reports of _cross-sectional studies_.**
(DOC)

## Acknowledgments

We would like to thank Imam Reza Hospital and Tabriz University of Medical Sciences staff for collaborating with the authors in conducting data collection.

## Author Contributions

**Conceptualization:** Habib Jalilian, Nazanin Mir.

**Data curation:** Habib Jalilian, Elnaz Javanshir, Nazanin Mir, Saeedeh Fehresti.

**Formal analysis:** Habib Jalilian.

**Investigation:** Habib Jalilian, Somayeh Heydari, Nazanin Mir.

**Methodology:** Habib Jalilian, Somayeh Heydari, Nazanin Mir.

**Project administration:** Habib Jalilian.

**Software:** Habib Jalilian.

**Supervision:** Habib Jalilian, Nazanin Mir.

**Validation:** Habib Jalilian, Nazanin Mir.

**Visualization:** Habib Jalilian.

**Writing – original draft:** Habib Jalilian, Somayeh Heydari, Elnaz Javanshir, Nazanin Mir, Saeedeh Fehresti.

**Writing – review & editing:** Habib Jalilian, Somayeh Heydari, Elnaz Javanshir, Khosro Jamebozorgi, Nazanin Mir, Abbas Eshraghi.

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
