## [Decision Letter · Decision Letter 0]

20 Nov 2023

PONE-D-23-22221Has the Health Sector Evolution Plan (HSEP) increased financial protection and reduced out-of-pocket (OOP) in lung cancer patients in Iran?PLOS ONE

Dear Dr. Mir,

Thank you for submitting your manuscript to PLOS ONE. After careful consideration, we feel that it has merit but does not fully meet PLOS ONE’s publication criteria as it currently stands. Therefore, we invite you to submit a revised version of the manuscript that addresses the points raised during the review process.

We look forward to receiving your revised manuscript.

Kind regards,

Masoud Behzadifar

Academic Editor

PLOS ONE

Journal Requirements:

Reviewers' comments:

Reviewer's Responses to Questions

**Comments to the Author**

1. Is the manuscript technically sound, and do the data support the conclusions?

Reviewer #1: Partly

Reviewer #2: Partly

2. Has the statistical analysis been performed appropriately and rigorously? 

Reviewer #1: Yes

Reviewer #2: Yes

3. Have the authors made all data underlying the findings in their manuscript fully available?

Reviewer #1: Yes

Reviewer #2: Yes

4. Is the manuscript presented in an intelligible fashion and written in standard English?

Reviewer #1: Yes

Reviewer #2: Yes

5. Review Comments to the Author

Reviewer #1: We thank the authors of the article for paying attention to the evaluations of the health system programs. This plan was one of the largest and most important programs of Iran's health system, which was implemented with the aim of increasing equity in health, and few studies have been conducted in the field of evaluating this plan, and the respected authors have answered many questions in this study, However, there are some points in my opinion that should be included in the article.

1-In the introduction of the abstract, instead of stating the result of the Health Sector Evolution Plan, the goals of this plan should be stated.

2-Keywords should be written based on Mesh Term

3-Explain the reason for using the bottom-up approach and the prevalence approach in the methodology.

4- Explain the relationship between financial protection and the use of a health system approach to cost estimation.

5-In the results, explain why the standard deviation of the costs is greater than their mean.

Reviewer #2: Dear Respectable Authors

Thank you for considering a great area of research related to cancer care. You investigated the impact of Health Sector Evolution Plan (HSEP) on financial protection and out-of-pocket (OOP) of lung cancer patients in Iran. Your results if of interest but the way you report the manuscript needs modification. I hope my comments will increase the quality of your manuscript.

- Title, It is better to state the title of the article in a non-question form. Ideally, the title of your research paper should be more informative so that it attracts the attention of the readers. It should provide more information about your research and the main outcome that you have achieved. It is not advisable to have a question as the title of your paper as it is the first thing readers will see about your paper. The aim of conducting research is to find answers and having a question in your title may not be attractive to the readers. However, there is no specific guideline about this and you may find a number of published articles that have questions as their titles.

- Affiliate numbers 2 and 3 are the same. Please refer to the journal format or ask editorial team to help you in this regard.

- Absract, please refine your subheading based on journal guideline. I prepared you the related link to the abstract section: https://plos.org/resource/how-to-write-a-great-abstract/

- As you stated in objective section of the abstract and also main text, one of your aims was to investigate "the impact of HSEP on hospitalization costs of lung cancer and patients’ contribution in Iran between 2010 and 2017". Please add the results related to this aim at the abstract section. In my opinion, you missed this important results here.

- Abstract, conclusion, considering journal guideline: "Tell the reader why your findings matter, and what this could mean for the ‘bigger picture’ of this area of research.", your conclusion is not based on your results. Please prepare a brief conclusion to the question of your study in simple and non-statistical language. for example, the HSEP increased financial protection and reduced OOP in lung cancer patients in Iran. Then, prepared one recommendation for future research based on your results, if possible.

- Conclusion, Your data collection period was between 2010-2017, how you conclude about post-2017 (line 64)? Please remove this statement.

- Please apply the corrections considered in the abstract to the main text as well.

- Introduction section, please remove subheading from the text.

- Which perspective do you use in your study. there is some difference between what stated in the abstract, line 52, and methods section, line 154. Please check and refine.

- In my opinion, it is better to write sample size as "statistical population". Please refine it in lines 48, and 159.

- Line 196, There is one additional point.

- Line 187 not in line with line 52. Two or Three? Please refine it and add the version for all three software in the abstract section.

- Line 221, Please add related SD.

- Please remove subheading from discussion section.

Cheers

-

6. PLOS authors have the option to publish the peer review history of their article (what does this mean?). If published, this will include your full peer review and any attached files.

Reviewer #1: No

Reviewer #2: No

---

## [Author Response · Author response to Decision Letter 0]

7 Jan 2024

Dear Editor,

Thank you very much for your insightful comments and suggestions. We have revised the manuscript, and would like to re-submit it for your consideration.

We have addressed the comments raised by the reviewers, and the amendments have been marked using Track Changes in the revised manuscript. Point by point responses to the reviewers’ comments are listed below this letter.

Kind regards,

Nazanin Mir

Reviewer #1: 

We thank the authors of the article for paying attention to the evaluations of the health system programs. This plan was one of the largest and most important programs of Iran's health system, which was implemented with the aim of increasing equity in health, and few studies have been conducted in the field of evaluating this plan, and the respected authors have answered many questions in this study, However, there are some points in my opinion that should be included in the article.

Dear Reviewer,

Thank you very much for providing us with the opportunity to revise and resubmit our manuscript. The insightful comments and suggestions have been immensely helpful. I hope we have addressed your concerns in the updated draft.

The amendments have been marked using Track Changes in the revised manuscript.

1-In the introduction of the abstract, instead of stating the result of the Health Sector Evolution Plan, the goals of this plan should be stated.

Response: Thank you for your valuable comment. Done. 

2-Keywords should be written based on Mesh Term

Response: Thank you for your valuable comment. Corrected. 

3-Explain the reason for using the bottom-up approach and the prevalence approach in the methodology.

Response: Thank you for your valuable comment.

1) The bottom-up approach is more precise to estimate the economic burden. Our data source was patients' medical records (medical billing), which is calculated by multiplying the number of services by unit costs (medical tariffs). The total cost is based on the sum of these costs. Hence, the bottom-up approach was used in this study. 

2) Our study's objective was not to evaluate lifetime cost. The timeframe of our study was one year (cost per hospitalization, drawing from the patient's medical record). So, the prevalence-based approach was used.

As you know:

In a bottom-up approach, the cost estimation can be stratified into two steps. The first step is to measure and quantify the health inputs employed and the second step is to estimate the unit costs of the inputs used to produce and confer specific medical and health care services. The total costs come out through the multiplication of unit costs by the quantities used. The top-down approach, known as the epidemiological or attributable risk approach, measures the proportion of a disease that is due to exposure to the disease or the risk factors.

Also, the COI studies can be described as prevalence-based or incidence-based approaches based on the way in which the epidemiological data are used. Being most commonly used, the former approach estimates the economic burden of a condition over a specific period, usually a year, while the latter approach estimates the lifetime costs of a condition from its onset until its disappearance (usually by cure or death), which refers to the new number of cases arising in a predefined time period. Prevalence-based studies estimate the number of cases of death and hospitalizations attributable to diseases in a given year and then estimate the costs that flow from those deaths or hospitalizations (plus other costs such as prevention, research and law enforcement costs). Incidence-based studies estimate the number of new cases of death or hospitalization in a given year and apply a lifetime cost estimate to these new cases.

Please see these articles: 

1) Jo C. Cost-of-illness studies: concepts, scopes, and methods. Clinical and molecular hepatology. 2014 Dec;20(4):327./ https://www.ncbi.nlm.nih.gov/pmc/articles/PMC4278062/pdf/cmh-20-327.pdf

2) Segel JE. Cost-of-illness studies—a primer. RTI-UNC center of excellence in health promotion economics. 2006 Jan;1:39./ https://www.researchgate.net/publication/253434922_Cost-of-Illness_Studies-A_Primer

4- Explain the relationship between financial protection and the use of a health system approach to cost estimation.

Response: Thank you for your valuable comment and point out our mistake.

 - Our study was conducted from the societal perspective. In this study, the total hospitalization costs were calculated. These costs comprise a portion from targeted subsidies, a portion from insurance and a portion from patient-copayment. Therefore, a societal perspective was utilized. 

5-In the results, explain why the standard deviation of the costs is greater than their mean.

Response: Thank you for your valuable comment.

-In this study a prevalence-based approach was used. In this approach, patients are selected from a period of time, as we did. These patients (included in our study) were in different stages of the disease, therefore, the variation in costs was high

Reviewer #2: 

Dear Respectable Authors

Thank you for considering a great area of research related to cancer care. You investigated the impact of Health Sector Evolution Plan (HSEP) on financial protection and out-of-pocket (OOP) of lung cancer patients in Iran. Your results if of interest but the way you report the manuscript needs modification. I hope my comments will increase the quality of your manuscript.

Dear Reviewer,

Thank you very much for providing us with the opportunity to revise and resubmit our manuscript. The insightful comments and suggestions have been immensely helpful. I hope we have addressed your concerns in the updated draft.

The amendments have been marked using Track Changes in the revised manuscript.

- Title, It is better to state the title of the article in a non-question form. Ideally, the title of your research paper should be more informative so that it attracts the attention of the readers. It should provide more information about your research and the main outcome that you have achieved. It is not advisable to have a question as the title of your paper as it is the first thing readers will see about your paper. 

The aim of conducting research is to find answers and having a question in your title may not be attractive to the readers. However, there is no specific guideline about this and you may find a number of published articles that have questions as their titles.

Response: Thank you for your valuable comment. Changed. 

-Hospitalization costs and out of pocket (OOP) payment in lung cancer patients in Iran: Health Sector Evolution Plan (HSEP) has reduced out-of-pocket (OOP) payments and improved financial protection 

- Affiliate numbers 2 and 3 are the same. Please refer to the journal format or ask editorial team to help you in this regard.

Response: Thank you for the valuable comment. Corrected.

- Abstract, 

please refine your subheading based on journal guideline. I prepared you the related link to the abstract section: https://plos.org/resource/how-to-write-a-great-abstract/

Response: Thank you for the valuable comment. This section has already been written according to the submission guidelines (authors’ guideline), If you have a specific point or particular matter in mind, please let us know.

- As you stated in objective section of the abstract and also main text, one of your aims was to investigate "the impact of HSEP on hospitalization costs of lung cancer and patients’ contribution in Iran between 2010 and 2017". Please add the results related to this aim at the abstract section. In my opinion, you missed this important results here.

Response: Thank you for the valuable comment. Corrected, (Page 3, Lines 110-116).

- Abstract, conclusion, considering journal guideline: "Tell the reader why your findings matter, and what this could mean for the ‘bigger picture’ of this area of research.", your conclusion is not based on your results. Please prepare a brief conclusion to the question of your study in simple and non-statistical language. for example, the HSEP increased financial protection and reduced OOP in lung cancer patients in Iran. Then, prepared one recommendation for future research based on your results, if possible.

Response: Thank you for your valuable comment, (Page 4, Lines 125-129).

- This study aimed to estimate the hospitalization costs and out-of-pocket payments before and after the HSEP. We aimed to see whether the hospitalization costs and patient contribution have changed. As our results (abstract section) highlighted, hospitalization costs had a rising trend and this trend continued after the HSEP, while OOP experienced a dropping trend since the HSEP.

It seems the HSEP increased financial protection and reduced OOP in lung cancer patients in Iran.

- Conclusion, Your data collection period was between 2010-2017, how you conclude about post-2017 (line 64)? Please remove this statement. Please apply the corrections considered in the abstract to the main text as well.

Response: Thank you for your comment and point out our mistake. Corrected.

- Introduction section, please remove subheading from the text.

Response: Thank you for the valuable comment. Done.

- Which perspective do you use in your study. there is some difference between what stated in the abstract, line 52, and methods section, line 154. Please check and refine.

Response: Thank you for your comment and point out our mistake. Corrected. This study was conducted from the societal perspective.

- In my opinion, it is better to write sample size as "statistical population". Please refine it in lines 48, and 159.

Response: Thank you for your comment and kind suggestion. Corrected. 

- Line 196, There is one additional point.

Response: Thank you for your comment and point out our mistake. Corrected. 

- Line 187 not in line with line 52. Two or Three? Please refine it and add the version for all three software in the abstract section.

Response: Thank you for your comment and point out our mistake. Corrected. 

- Line 221, Please add related SD.

Response: Thank you for the valuable comment. Added. 

- Please remove subheading from discussion section.

Response: Thank you for the valuable comment. Done.

Cheers

---

## [Decision Letter · Decision Letter 1]

16 Jan 2024

Hospitalization costs and out of pocket (OOP) payment in lung cancer patients in Iran: Health Sector Evolution Plan (HSEP) has reduced out-of-pocket (OOP) payments and improved financial protection

PONE-D-23-22221R1

Dear Dr. Mir,

We’re pleased to inform you that your manuscript has been judged scientifically suitable for publication and will be formally accepted for publication once it meets all outstanding technical requirements.

Kind regards,

Masoud Behzadifar

Academic Editor

PLOS ONE

Additional Editor Comments (optional):

Reviewers' comments:

Reviewer's Responses to Questions

**Comments to the Author**

1. If the authors have adequately addressed your comments raised in a previous round of review and you feel that this manuscript is now acceptable for publication, you may indicate that here to bypass the “Comments to the Author” section, enter your conflict of interest statement in the “Confidential to Editor” section, and submit your "Accept" recommendation.

Reviewer #2: All comments have been addressed

2. Is the manuscript technically sound, and do the data support the conclusions?

Reviewer #2: Yes

3. Has the statistical analysis been performed appropriately and rigorously? 

Reviewer #2: Yes

4. Have the authors made all data underlying the findings in their manuscript fully available?

Reviewer #2: Yes

5. Is the manuscript presented in an intelligible fashion and written in standard English?

Reviewer #2: Yes

6. Review Comments to the Author

Reviewer #2: Dear respected Authors

Thank you for your clarification. In my opinion, your manuscript is acceptable in this fashion.

Cheers

7. PLOS authors have the option to publish the peer review history of their article (what does this mean?). If published, this will include your full peer review and any attached files.

Reviewer #2: No

---

## [Editor Report · Acceptance letter]

15 Aug 2024

PONE-D-23-22221R1 

PLOS ONE

Dear Dr. Mir, 

I'm pleased to inform you that your manuscript has been deemed suitable for publication in PLOS ONE. Congratulations! Your manuscript is now being handed over to our production team.

Kind regards, 

on behalf of

Dr. Masoud Behzadifar 

Academic Editor

PLOS ONE